# FITTING FEATURE NORM TO CONFIDENCE: A REGULARIZATION APPROACH FOR ROBUST OUT-OF-DISTRIBUTION DETECTION

## ABSTRACT

We propose a novel framework for robust out-of-distribution (OOD) detection by explicitly designing the feature space. Our approach aligns feature norm with model confidence by enforcing a zero-confidence baseline—defined as the feature norm at which the softmax output is uniform—and deriving an upper bound on the feature norm through softmax sensitivity analysis. This strategy enables in-domain samples to exhibit high confidence while ensuring that OOD samples, which naturally possess lower feature norms, yield near-uniform predictions. Unlike existing methods that simply modify the feature norm without optimizing the underlying embedding space, our method learns an optimal feature space via density ratio estimation using Kernel Logistic Regression and feature space augmentation. Our theoretical analysis shows that the risk difference between the true data distribution (comprising both known and unknown samples) and an auxiliary domain—constructed from augmented OOD samples drawn from the inner region of the feature space—is bounded. Extensive experiments show that our approach significantly enhances OOD detection performance.

## 1 INTRODUCTION

In recent years, various tasks such as Positive-Unlabeled (PU) learning (Garg et al., 2021; Bekker & Davis, 2020; Kiryo et al., 2017), open set recognition (Vaze et al., 2022), uncertainty estimation (Lakshminarayanan et al., 2017), and out-of-distribution (OOD) detection (Liu et al., 2020; Hendrycks & Gimpel, 2017; Dong et al., 2022) have emerged as critical challenges in machine learning. Each task addresses a distinct aspect of model performance under varying conditions, from handling unknown classes to predicting system failures. Although many researchers have developed tailored solutions for these problems, most methods treat them separately and often overlook the underlying structure of the feature space.

A common strategy is to use the feature norm as an indicator of confidence. In standard classification networks, the logit for class $i$ is calculated as $z_i = w_i^\top x = \|w_i\|\|x\|\cos(\theta_i)$, where $x$ denotes the embedding of the input, $w_i$ is the corresponding weight vector (or class proxy), and $\cos(\theta_i)$ measures the angular similarity between $x$ and $w_i$. Intuitively, a lower feature norm $\|x\|$ should imply lower confidence. However, due to the exponential transformation in the softmax function, even small increases in $\|x\|$ can lead to disproportionate changes in predicted probability, especially in high-dimensional spaces where cosine similarities are highly concentrated (peaky).

To address this issue, we propose a novel OOD detection framework that explicitly aligns feature norm with confidence. Our key insight is that if feature norm and confidence are well-linked, OOD detection can be easily facilitated. Specifically, we define a *zero-confidence* baseline—namely, the feature norm at which the softmax output becomes uniform (*i.e.*, $1/K$ for $K$ classes)—and derive an upper bound on the feature norm by analyzing the sensitivity of the softmax mapping. This upper bound ensures that further increases in the feature norm yield only moderate changes in confidence. By generating OOD embeddings within a controlled region of the embedding space—determined by the ratio of the zero-confidence baseline to the upper bound—and enforcing that these augmented samples yield uniform predictions, we encourage in-distribution (ID) samples to remain within a narrow, interpretable range of feature norms. In turn, OOD samples, which are naturally associated with low feature norms, yield near-uniform softmax outputs.

Unlike previous methods (Park et al., 2023) that modify the feature norm for OOD detection while retaining standard classification loss, our approach optimizes the feature space to establish a tight link between the feature norm and confidence. We achieve this by employing density ratio estimation techniques such as Kernel Logistic Regression and validating our strategy through feature space augmentation. Our theoretical analysis demonstrates that the risk difference between the true domain—*i.e.*, the overall data distribution containing both known and unknown samples—and an auxiliary domain, constructed using augmented OOD samples drawn from the inner region of the feature space, is bounded. This bound suggests that our augmented OOD samples may reasonably approximate the unknown domain, indicating that when these samples lie within the controlled region, they can serve a similar role to actual OOD data in supporting robust OOD detection.

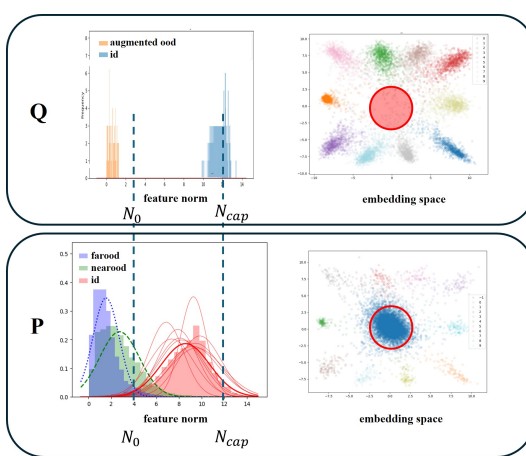

Figure 1: **Comparison of the auxiliary domain $Q$ (augmented OOD samples) with the true distribution $P$.** Left: Feature norm distributions. Right: Embedding space visualization, showing the region (marked $N_0$) where augmented OOD samples approximate actual OOD data.

Fig.1 compares the auxiliary domain $Q$, which contains the augmented OOD samples, with the true distribution $P$ that includes the actual OOD data. The first column shows the distribution of feature norms, and the second column depicts the embedding space. In the feature norm plot, we observe that through our proposed loss and training, the distribution of feature norms for augmented OOD samples in $Q$ closely matches that of the actual OOD dataset in $P$. Furthermore, in the embedding space, the region occupied by the augmented OOD samples within $N_0$ is similar to the area where the actual OOD data is located, with the red circle indicating $N_0$. Thus, this bound demonstrates that our augmented OOD samples effectively approximate the unknown domain, ensuring that when these samples lie within the controlled region, they play a role similar to actual OOD data in driving robust OOD detection.

Our extensive experiments confirm that our adaptive regularization significantly enhances OOD detection performance. By unifying these ideas within an optimized feature space, our work provides a strong theoretical foundation for future research in feature space optimization.

## 2 RELATED WORK

**Recent OOD Detection Technique.** Recent advances in OOD detection, uncertainty estimation, and confidence calibration have produced a rich body of research addressing the challenge of reliably distinguishing ID from OOD samples. Early efforts include dropout-based uncertainty estimation (Gal & Ghahramani, 2016), which treated dropout as an approximate Bayesian inference method to explore model uncertainty. In a complementary approach, the trust score method (Jiang et al., 2018) investigated the correlation between classifier outputs and prediction errors. Furthermore, several studies (Liang et al., 2018; Hendrycks & Gimpel, 2017) established baselines for OOD detection based on the maximum softmax probability (MSP), showing that misclassified and OOD samples tend to exhibit lower MSP values. Subsequent research (Hein et al., 2019) has analyzed the inherent overconfidence of deep networks on OOD inputs and proposed mitigation strategies.

More recent methods leverage auxiliary data and tailored loss functions to further improve OOD detection. For instance, OE (Hendrycks et al., 2019a) employed external outlier data to refine decision boundaries, while ensemble techniques and self-supervised leave-out classifiers have also been explored (Vyas et al., 2018). Additionally, ReweightOOD (Regmi et al., 2024b) addressed challenges arising from overlapping ID and OOD regions in the embedding space by adaptively reweighting samples during contrastive optimization. It classified samples as easy or hard positives/negatives based on their proximity to class centroids and improved separability.

Our approach differs from these methods as it does not rely on external OOD data or ensembles. Instead, we construct an auxiliary domain bounded by the true joint distribution and generate augmented OOD samples directly in the feature space. This enables robust OOD detection without sacrificing the simplicity of the training objective.

**Feature Norm-Based Approaches.** Several recent studies have examined the role of the feature norm in the embedding space. In face recognition applications, MagFace (Meng et al., 2021) has shown that the feature norm correlates with image quality. Moreover, other studies (Yu et al., 2020; Ranjan et al., 2017) observed that ID samples tend to exhibit larger embedding magnitudes than OOD samples. Dhamija et al. (2018) employed an entropic open-set loss to reduce the feature norm of unknown samples, thereby enhancing the separability between known and unknown classes. Additionally, studies in open-set recognition (Vaze et al., 2022) have demonstrated that during training, the feature norms of known classes increase, while uncertain or unknown samples remain close to the origin. Most relevant to our work (Park et al., 2023) theoretically established that feature norm is equivalent to a classifier's confidence, supporting OOD detection via confidence-based separation. Building on these insights, T2FNorm (Regmi et al., 2024a) enforced hyperspherical constraints on ID embeddings during training so that at test time, weakly activated OOD samples exhibit lower norms and can be separated from ID samples by thresholding the maximum softmax probability.

In contrast, our approach directly aligns feature norm with confidence by defining a zero-confidence baseline (*i.e.*, the feature norm at which the softmax output is uniform) and deriving an upper bound on the feature norm based on the sensitivity of the softmax mapping. We then employ a targeted augmentation strategy to generate OOD samples within a controlled, low-confidence region. This explicit calibration of the feature space ensures that our augmented OOD samples approximate the unknown domain, enabling robust OOD detection without additional density estimation modules.

## 3 METHOD

Let $X$ be a random variable representing input images and $Y$ be the corresponding label. Here, $Y$ is a discrete random variable taking values in the set $\{1, \ldots, K\}$, corresponding to a multi-class classification task with $K$ classes. We assume that the true joint distribution of $X$ and $Y$ is given by $p_{\text{data}}$. From this distribution, we obtain $N$ independent and identically distributed (i.i.d.) samples, forming the dataset $\mathcal{D} = \{(x_n, y_n)\}_{n=1}^N$. The function $f$ denotes a deep neural network trained for multi-class classification. Given an input $x \in X$, the network $f$ produces two outputs: the predicted label $\hat{Y}$ and the logit vector $\hat{z}$. The logits $\hat{z}$ are subsequently passed through the softmax function $\sigma_{SM}$ to convert them into a probabilistic vector $\hat{P}$. This probabilistic output $\hat{P}$ is referred to as the confidence score, representing the network's confidence in its prediction.

A critical component in the network is its final fully connected layer, which maps the embedding space $o$ (typically the output of one of the last hidden layers of $f$) to the logit space. Let $g : \mathcal{X} \to \mathbb{R}^d$ denote the embedding function that maps an input $x$ to an embedding $g(x)$, and let $h : \mathbb{R}^d \to \mathbb{R}^K$ denote the mapping from the embedding space to the logit space. This layer is parameterized by the weight matrix $W$ and plays a crucial role in transforming the learned embeddings $o$ into logits $\hat{z}$, which are then used to determine the final classification output.

### 3.1 LEARNING FOR CONFIDENCE-BASED OOD DETECTION

**Definition 1** (Domain). Given an input space $\mathcal{X}$ and a label space $\mathcal{Y}$, a domain is defined as any joint distribution $P_{X,Y}$ over $\mathcal{X} \times \mathcal{Y}$. In this context, the known classes form a subset of $\mathcal{Y}$, denoted as $\mathcal{Y}_k$. The unknown classes are defined as those belonging to the set $\mathcal{Y}_u = \mathcal{Y} \setminus \mathcal{Y}_k$.

We aim to train a classifier $f$ using samples $\mathcal{D} = \{(x_i, y_i)\}_{i=1}^n$ drawn from the in-distribution $P_{X,Y|Y \in \mathcal{Y}_k}$ so that $f$ can correctly classify ID samples and reliably identify OOD samples by producing near-uniform softmax outputs for unknown inputs.

**Problem 1** (Learning for Reliable Confidence-based OOD Detection). *Given independent and identically distributed (i.i.d.) samples $\mathcal{D} = \{(x_i, y_i)\}_{i=1}^n$ drawn from $P_{X,Y|Y \in \mathcal{Y}_k}$, the goal of learning for reliable confidence-based OOD detection is to train a classifier $f$ using $\mathcal{D}$ such that $f$ can:*

1. *correctly classify samples from known classes,*

2. *identify samples from unknown classes, and*

3. *produces near-uniform predictions for OOD samples, thereby indicating low confidence.*

These objectives correspond to different aspects of the task: the first condition addresses the classification in a closed-set scenario, the second focuses on OOD detection, and the third relates to reliable confidence (*i.e.*, ensuring that the model is less confident when faced with unknown samples). From the perspective of statistical learning, we formalize this by defining risks over the logit space.

**Logit Space Risk.** Let $l : \mathbb{R}^K \times \mathbb{R}^K \to \mathbb{R}_{\geq 0}$ be a loss function, and let $h$ represent any hypothesis function in the set $\{h : \mathbb{R}^D \to \mathbb{R}^K\}$ and any hypothesis function $g$ in the set $\{g : \mathcal{X} \to \mathbb{R}^D\}$. For the known classes, the logit risk is defined in the standard manner as:

$$R_{P,k}(h) := \int_{\mathcal{X} \times \mathcal{Y}_k} l\Big(h\big(g(x)\big), y\Big) \, dP_{X,Y|Y \in \mathcal{Y}_k}(x, y). \tag{1}$$

For the unknown classes, where ground-truth labels are unavailable, we adopt a uniform target distribution. Specifically, for a classification task with $K$ known classes, we define the uniform target $\bar{\boldsymbol{y}} \in \mathbb{R}^K$ as

$$\bar{\boldsymbol{y}} = \left(\frac{1}{K}, \frac{1}{K}, \dots, \frac{1}{K}\right). \tag{2}$$

This uniform target represents complete uncertainty among the $K$ classes and encourages the classifier to output a flat probability distribution for unknown inputs. Thus, the logit risk for unknown classes is defined as

$$R_{P,u}(h) := \int_{\mathcal{X}} l\Big(h\big(g(x)\big), \bar{\boldsymbol{y}}\Big) \, dP_{X|Y \in \mathcal{Y}_u}(x). \tag{3}$$

These two risk components (logit space risk for both known and unknown classes) directly support the key objectives of our framework: ensuring accurate in-domain classification, robust OOD detection, and reliable confidence calibration. Specifically, risks associated with known classes promote high confidence in correct predictions, while the risk associated with unknown classes enforces low confidence (*i.e.*, near-uniform predictions) for OOD samples.

The total risk for the distribution $P_{X,Y}$ can be formulated as a weighted combination of the risks for known and unknown classes. In the logit space, the $\alpha$-risk is defined as:

$$R_P^\alpha(h) := (1 - \alpha)R_{P,k}(h) + \alpha R_{P,u}(h), \tag{4}$$

where $\alpha$ is a weighting factor that reflects the importance of the unknown classes.

## 3.2 AUXILIARY DOMAIN CONSTRUCTION

Since no samples are available for unknown classes, directly computing the partial risk for these classes is challenging. To address this, we introduce an auxiliary domain $Q_{X,Y}$, following a similar approach to that in Fang et al. (2021).

**Definition 2** (Auxiliary Domain). A domain $Q_{X,Y}$ defined over $\mathcal{X} \times \mathcal{Y}$ is called the auxiliary domain for $P_{X,Y}$ if it satisfies the following conditions: $Q_{X|Y \in \mathcal{Y}_k} = P_{X|Y \in \mathcal{Y}_k}$, $Q_{Y|X} = P_{Y|X}$, and $P_X \ll Q_X$, where $P_X \ll Q_X$ denotes absolute continuity.

**Theorem 1** (Boundedness of the Logit $\alpha$-Risk Difference). *Given a loss function $\ell$ that satisfies the triangle inequality, and a hypothesis space $\mathcal{H} \subset \{\boldsymbol{h} : \mathcal{X} \to \mathbb{R}^K\}$, if $Q_{X,Y}$ is the auxiliary domain for $P_{X,Y}$ (with the definitions of $R_{P,u}(\boldsymbol{h})$, $R_{Q,u}(\boldsymbol{h})$ and $R_{P,k}(\boldsymbol{h}) = R_{Q,k}(\boldsymbol{h})$), then for any $\boldsymbol{h} \in \mathcal{H}$, the following bound holds:*

$$\left| R_P^\alpha(h) - R_Q^\alpha(h) \right| \leq \alpha \, d_{h,\mathcal{H}}^l\Big(P_{X|Y \in \mathcal{Y}_u}, Q_{X|Y \in \mathcal{Y}_u}\Big) + \alpha \, \Lambda, \tag{5}$$

*where the disparity discrepancy metric is defined as*

$$d_{h,\mathcal{H}}^l\Big(P_{X|Y \in \mathcal{Y}_u}, Q_{X|Y \in \mathcal{Y}_u}\Big) := \sup_{h' \in \mathcal{H}} \left| \int_{\mathcal{X}} \ell\Big(h'(g(x)), \bar{y}\Big) \, d\Big(P_{X|Y \in \mathcal{Y}_u} - Q_{X|Y \in \mathcal{Y}_u}\Big)(x) \right|, \tag{6}$$

*and the residual term is defined as*

$$\Lambda := \min_{h' \in \mathcal{H}} \Big(R_{P,u}(h') + R_{Q,u}(h')\Big). \tag{7}$$

*Furthermore, we prove that*

$$\min_{h \in \mathcal{H}} R_Q^\alpha(h) = \min_{h \in \mathcal{H}} R_P^\alpha(h) \;\; and \;\; \arg\min_{h \in \mathcal{H}} R_Q^\alpha(h) \subset \arg\min_{h \in \mathcal{H}} R_P^\alpha(h). \tag{8}$$

*That is, the hypothesis that minimizes the risk in the auxiliary domain $Q$ minimizes the risk in the true domain $P$.*

*Proof.* Please refer to Appendix A. $\qquad\square$

To effectively handle unknown classes without available samples, we propose constructing an auxiliary domain $Q_{X,Y}$. This auxiliary domain is designed to closely approximate the overall data distribution by incorporating both known and unknown samples. To achieve this, we first define a latent space distribution $U$ that uniformly covers the range of the embeddings. Let $Z \subset \mathbb{R}^d$ denote the latent (embedding) space produced by the network, and let $g : \mathcal{X} \rightarrow \mathbb{R}^d$ be the embedding function. We define $U$ as a probability distribution over $Z$, specifically as the uniform distribution over the hyper-rectangle:

$$U = \prod_{j=1}^{d} [m_j, M_j], \tag{9}$$

where, for each dimension $j$,

$$m_j = \min_{x \in \mathcal{D}} g_j(x) \quad \text{and} \quad M_j = \max_{x \in \mathcal{D}} g_j(x). \tag{10}$$

Thus, $U$ uniformly covers the range of the embedding space induced by the training (ID) dataset $\mathcal{D}$.

Following Fang et al. (2021), we assume that the known class conditional distribution $P_{X|Y \in \mathcal{Y}_k}$ is absolutely continuous with respect to $U$; that is, for any $U$-measurable set $A \subset Z$,

$$P_{X|Y \in \mathcal{Y}_k}(A) = \int_A r(x) \, dU(x), \tag{11}$$

where $r(x) = \frac{dP_{X|Y \in \mathcal{Y}_k}(x)}{dU(x)}$ is the density ratio.

We define $S$ as the set of latent representations of ID samples (*i.e.*, drawn from $P_{X,Y|Y \in \mathcal{Y}_k}$) and $T$ as the set of samples drawn from the uniform distribution $U$ over the latent space. To estimate the density ratio, we train a kernel logistic regression (KLR) model. In our formulation, samples from $S$ serve as positive examples, while samples from $T$ are treated as unlabeled. Specifically, the KLR estimates

$$w(z) = 1/1 + \exp(-\phi(z)), \tag{12}$$

where $z$ is the latent representation and we set $\phi(z) = \|x\|$. Hence, the weight $w(z)$ reflects the likelihood that a sample from $T$ is in-domain.

Leveraging this analysis, we simplify the auxiliary domain construction. Rather than performing full density estimation over the entire embedding space, we treat samples from $U$ with feature norms below a specified threshold as OOD, while directly using the actual ID samples. This yields an auxiliary domain consisting of true in-domain data and augmented OOD samples (uniformly sampled from the low feature norm region), approximating the unknown domain without complex density estimation procedures.

### 3.3 Feature Norm-Based Augmentation and Learning Dynamics

Our approach leverages the intrinsic structure of the embedding space to generate augmented OOD samples and regulates the mapping between the feature norm and the network output. As discussed in Introduction, when the feature norm exceeds a certain threshold, the softmax output tends to saturate, which can disrupt the reliable linking of feature norm to confidence. To maintain a strong and consistent mapping between feature norm and confidence, we identify a controlled range where this mapping is smooth. This process is driven by two key thresholds computed from ID data:

• *Confidence Baseline Norm ($N_0$):* This is the feature norm at which the softmax output becomes uniform (*i.e.*, each class receives a probability of $1/K$), corresponding to a state of zero confidence.

• *Confidence Saturation Bound ($N_{\text{cap}}$):* This is defined as the smallest feature norm above $N_0$ at which the numerical derivative of the softmax output with respect to the feature norm exceeds a specified threshold (*e.g.*, 1.0). In other words, $N_{\text{cap}}$ marks the upper limit of the region where the mapping from feature norm to softmax output remains smooth and controlled. In our framework, ID samples are encouraged to have feature norms below $N_{\text{cap}}$ to ensure a reliable link between the feature norm and the model's confidence.

We then define a scaling ratio $\tau = \frac{N_0}{N_{\text{cap}}}$, which is used to guide our OOD augmentation.

**Forward Pass and Threshold Computation.** During a forward pass, the network produces embeddings $g(x)$. We compute the feature norm $\|g(x)\|$ and the cosine similarities between $g(x)$ and each class representative. These statistics are used to determine $N_0$ by identifying the candidate feature norm at which the softmax output is closest to uniform (*i.e.*, $1/K$). We then obtain $N_{\text{cap}}$ as the smallest candidate feature norm above $N_0$ at which the approximate (numerical) derivative of

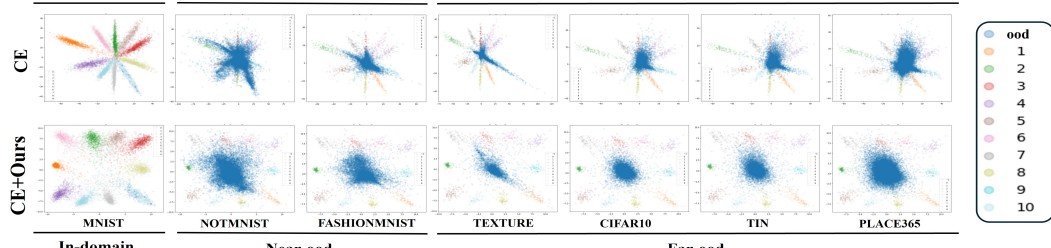

Figure 2: **Visualization of the feature space for the baseline Cross Entropy (CE) method (top row) and our proposed CE+Ours method (bottom row)** across various datasets. The proposed method shows improved feature space organization, with clusters positioned further from the origin, indicating larger feature norms. Consequently, the OOD samples from six datasets are concentrated near the origin, forming a balanced circular distribution with smaller feature norms. In contrast, the baseline CE method exhibits clustering bias, with OOD samples more unevenly distributed.

Table 1: **Performance comparison between the Cross Entropy (CE) method and the CE+Ours method** on various datasets for OOD detection tasks in terms of FPR@95, AUROC, AUPR-in, and AUPR-out. Notably, the accuracy for CE and CE+Ours is 95.81% and 95.14%, respectively.

| Dataset | Method | FPR@95↓ | AUROC↑ | AUPR-in↑ | AUPR-out↑ |
|---|---|---|---|---|---|
| NotMNIST | CE / CE+Ours | 87.24 / **16.20** | 81.45 / **96.40** | 60.34 / **93.84** | 88.68 / **97.88** |
| FashionMNIST | CE / CE+Ours | 21.91 / **9.65** | 93.06 / **97.80** | 92.82 /**97.89** | 92.03 / **97.62** |
| NearOOD | CE / CE+Ours | 54.58 /**12.93** | 87.25 / **97.10** | 76.58 / **95.89** | 90.36 / **97.62** |
| Texture | CE / CE+Ours | 57.90 / **6.42** | 88.97 / **98.55** | 87.77 / **98.74** | 86.25 /**98.00** |
| CIFAR-10 | CE/ CE+Ours | 23.83 / **2.53** | 93.79 /**99.30** | 93.57 / **99.34** | 93.62 /**99.26** |
| TinyImageNet | CE / CE+Ours | 26.97 / **3.64** | 93.13 / **99.01** | 92.58 /**98.96** | 93.05 / **99.01** |
| Places365 | CE | 24.76 / **3.82** | 93.53 / **99.00** | 82.50 / **97.13** | 97.96 /**99.71** |
| FarOOD | CE | 33.36 / **4.10** | 92.36 / **98.96** | 89.11 / **98.54** | 92.72 / **98.99** |

the softmax output with respect to the feature norm exceeds a specified threshold $\delta$ (*e.g.*, 1.0). Their ratio, $\tau$, is subsequently calculated.

**OOD Augmentation.** For each batch, we compute the average ID feature norm, $\mu_{ID}$. We then generate augmented OOD embeddings by uniformly sampling within a ball of radius $\tau \cdot \mu_{ID}$ in the embedding space. This random augmentation ensures that the generated OOD samples lie within a controlled, low-confidence region—effectively approximating the unknown domain.

**Learning Dynamics.** The final training objective is formulated as a weighted combination of two loss terms: 1.ID Loss: The standard cross-entropy loss computed on ID samples, which promotes correct classification. 2.OOD Loss: A loss computed on the augmented OOD embeddings that penalizes high-confidence outputs (*e.g.*, by reducing the maximum softmax probability).

If an ID sample's feature norm becomes excessively large, its OOD augmentation (scaled by $\tau$) incurs a penalty that counteracts further norm increases. This mechanism enforces a smooth and continuous mapping between feature norm and network output, ensuring that ID samples maintain high confidence while augmented OOD samples yield near-uniform (low-confidence) predictions.

In summary, by explicitly computing $N_0$ (the zero-confidence baseline) and $N_{cap}$ (the confidence saturation bound) to derive the scaling ratio $\tau = \frac{N_0}{N_{cap}}$, our method employs feature norm-based augmentation to generate OOD samples in a controlled, low-confidence region of the embedding space. This strategy ensures that ID samples remain in a range where small changes in norm yield gradual changes in output, while augmented OOD samples approximate the unknown domain. By relying solely on intrinsic feature norms without additional density estimation modules, our approach naturally aligns feature norms with network confidence. Combined with a joint training loss, this results in an optimal embedding space where the risk difference between the true distribution and the augmented domain is bounded, ensuring robust OOD detection.

## 4 EXPERIMENTS

For our experiments, we followed the OpenOOD benchmark (Yang et al., 2022), which emphasizes fair comparison in OOD detection tasks. OpenOOD addresses several key challenges in the field: (1) Although OOD detection shares a common goal with tasks like open-set learning, open-set recognition, and novelty detection, these tasks are often referred to by different names, leading to inconsistency in evaluation. (2) Due to the nature of the task, experiments require sepa-

Table 2: **FPR@95 results on CIFAR-10** with various OOD sets. Lower FPR@95 values indicate better performance. The best and second-best values are boldfaced and underlined, respectively.

| Training | Post-processor | Near-OOD | | | Far-OOD | | | | | ID ACC |
|---|---|---|---|---|---|---|---|---|---|---|
| | | CIFAR-100 | TIN | *Avg.* | MNIST | SVHN | Textures | Places365 | *Avg.* | |
| ConfBranch(DeVries & Taylor, 2018) | ConfBranch | 34.44 | 28.11 | 31.28 | 15.79 | 14.06 | 27.24 | 28.85 | 21.48 | 94.88 |
| RotPred(Hendrycks et al., 2019b) | RotPred | 34.24 | 22.04 | 28.14 | 9.24 | 3.20 | 9.87 | 26.61 | 12.23 | 95.35 |
| G-ODIN(Hsu et al., 2020) | G-ODIN | 48.86 | 42.21 | 45.54 | 4.53 | 10.72 | 27.27 | 43.30 | 21.45 | 94.70 |
| CSI(Tack et al., 2020) | MSP | 37.57 | 29.74 | 33.66 | 24.41 | 17.56 | 28.95 | 34.76 | 26.42 | 91.16 |
| ARPL(Chen et al., 2021) | ARPL | 43.38 | 37.28 | 40.33 | 21.49 | 35.68 | 35.19 | 37.21 | 32.39 | 93.66 |
| MOS(Huang & Li, 2021) | MOS | 79.38 | 78.05 | 78.72 | 65.95 | 57.79 | 76.78 | 51.09 | 62.90 | 94.83 |
| VOS(Du et al., 2022) | EBO | 61.57 | 52.49 | 57.03 | 35.92 | 31.50 | 46.53 | 47.78 | 40.43 | 94.31 |
| LogitNorm(Wei et al., 2022) | MSP | 34.37 | 24.30 | 29.34 | 3.93 | 8.33 | 21.94 | 21.04 | 13.81 | 94.30 |
| CIDER(Ming et al., 2023) | KNN | 35.60 | 28.61 | 32.11 | 24.76 | 8.04 | 25.05 | 25.03 | 20.72 | - |
| NPOS(Tao et al., 2023) | KNN | 35.71 | 29.57 | 32.64 | 22.96 | 6.41 | 20.80 | 32.19 | 20.59 | - |
| T2FNorm(Regmi et al., 2024a) | T2FNorm | 30.60 | 22.33 | 26.47 | 3.50 | 5.72 | 19.49 | 22.27 | 12.75 | 94.69 |
| OE(Hendrycks et al., 2019a) | MSP | 36.71 | **2.97** | 19.84 | 24.67 | **1.25** | 12.07 | 14.53 | 13.13 | 94.63 |
| MCD(Yu & Aizawa, 2019) | MCD | 34.36 | 25.98 | 30.17 | 62.11 | 19.43 | 22.51 | 24.10 | 32.03 | 94.95 |
| UDG(Yang et al., 2021) | MSP | 40.75 | 29.93 | 35.34 | 16.61 | 17.39 | 19.70 | 27.70 | 20.35 | 92.36 |
| MixOE(Zhang et al., 2023) | MSP | 58.29 | 44.62 | 51.45 | 38.28 | 20.36 | 33.19 | 43.54 | 33.84 | 94.55 |
| RandAugment(Cubuk et al., 2020) | MSP | 40.08 | 30.95 | 35.51 | 15.03 | 20.97 | 30.30 | 30.33 | 24.16 | 95.59 |
| AugMix(Hendrycks et al., 2020) | MSP | 41.83 | 33.52 | 37.68 | 23.74 | 24.42 | 25.61 | 34.22 | 27.00 | 95.01 |
| PixMix(Hendrycks et al., 2022) | MSP | 28.91 | 21.01 | 24.96 | 27.44 | 10.14 | 6.64 | 21.11 | 16.33 | 95.15 |
| RegMixup(Pinto et al., 2022) | MSP | 54.89 | 42.68 | 48.78 | 20.74 | 36.94 | 44.30 | 43.21 | 36.30 | **95.75** |
| **Ours** | MSP | **25.31** | 20.26 | **22.79** | **2.75** | 3.62 | **5.62** | **11.63** | **5.91** | 95.32 |

Table 3: **AUROC on CIFAR-10** with OOD sets. Higher values indicate better performance.

| *Alg.* | Near-OOD | | | Far-OOD | | | | |
|---|---|---|---|---|---|---|---|---|
| | CIFAR-100 | TIN | *Avg.* | MNIST | SVHN | Textures | Places365 | *Avg.* |
| DeVries & Taylor (2018) | 88.91 | 90.77 | 89.84 | 94.49 | 95.42 | 91.10 | 90.39 | 92.85 |
| Hendrycks et al. (2019b) | 91.19 | 94.17 | 92.68 | 97.52 | 98.89 | 97.30 | 92.76 | 96.62 |
| Hsu et al. (2020) | 88.14 | 90.09 | 89.12 | 98.95 | 97.76 | 95.02 | 90.31 | 95.51 |
| Tack et al. (2020) | 88.16 | 90.87 | 89.51 | 92.55 | 95.18 | 90.71 | 89.56 | 92.00 |
| Chen et al. (2021) | 86.76 | 88.12 | 87.44 | 92.62 | 87.69 | 88.57 | 88.39 | 89.31 |
| Huang & Li (2021) | 70.57 | 72.34 | 71.45 | 74.81 | 73.66 | 70.35 | 86.81 | 76.41 |
| Du et al. (2022) | 86.57 | 88.84 | 87.70 | 91.56 | 92.18 | 89.68 | 89.90 | 90.83 |
| Wei et al. (2022) | 90.95 | 93.70 | 92.33 | 99.14 | 98.25 | 94.77 | 94.79 | 96.74 |
| Ming et al. (2023) | 89.47 | 91.94 | 90.71 | 93.30 | 98.06 | 93.71 | 93.77 | 94.71 |
| Tao et al. (2023) | 88.57 | 90.99 | 89.78 | 92.64 | 98.88 | 94.44 | 90.32 | 94.07 |
| Regmi et al. (2024a) | 91.56 | 94.02 | 92.79 | 99.28 | 98.81 | 95.44 | 94.40 | 96.98 |
| Hendrycks et al. (2019a) | 90.54 | 99.11 | 94.82 | 90.22 | 99.60 | 97.58 | 96.58 | 96.00 |
| Yu & Aizawa (2019) | 89.88 | 92.18 | 91.03 | 84.22 | 93.76 | 93.35 | 92.66 | 91.00 |
| Yang et al. (2021) | 88.62 | 91.20 | 89.91 | 95.81 | 94.55 | 93.92 | 91.97 | 94.06 |
| Zhang et al. (2023) | 87.47 | 90.00 | 88.73 | 91.66 | 93.82 | 91.84 | 90.38 | 91.93 |
| Cubuk et al. (2020) | 89.26 | 91.03 | 90.15 | 95.26 | 93.33 | 91.17 | 91.12 | 92.72 |
| Hendrycks et al. (2020) | 88.61 | 90.26 | 89.43 | 92.33 | 92.19 | 91.91 | 90.19 | 91.66 |
| Hendrycks et al. (2022) | 90.86 | 92.65 | 91.76 | 98.47 | 99.18 | **98.27** | **98.09** | 98.50 |
| Pinto et al. (2022) | 84.71 | 85.96 | 85.33 | 99.02 | 94.94 | 81.66 | 96.19 | 92.95 |
| **Ours** | **91.82** | 94.88 | 93.35 | 99.23 | 98.96 | 98.56 | 97.48 | **98.56** |

rate in-domain and OOD domains. However, the choice of datasets for each domain lacks consistency across studies. Thus, OpenOOD defines both near-OOD and far-OOD categories relative to the training data. `Appendix C and D presents more results on ImageNet-200 and outlines our hyperparameter tuning strategy.`

### 4.1 EXPERIMENTS ON MNIST

We conducted the experiments on the MNIST dataset. In the OpenOOD framework, near-OOD datasets include notMNIST and FashionMNIST, while far-OOD datasets consist of Texture, CIFAR-10, TinyImageNet (TIN), and Places365. For evaluation, we used the following metrics: classification accuracy, FPR@95, AUROC, AUPR-in, and AUPR-out, reporting the average performance across both near-OOD and far-OOD scenarios. For the network architecture, we used a LeNet model, setting the dimensionality of the pre-softmax layer to 2 for visualization purposes. To utilize all regions in the embedding space, the final layer ReLU activation was omitted. Additionally, we employed Maximum Softmax Probability (MSP) to perform OOD evaluation across both the baseline model and our method (CE+Ours). We used standard MSP to align with recent training-time methods that avoid post-hoc detectors. The hyperparameters are set to $\alpha = 0.1$ and $\delta = 1.0$.

Table 1 shows that our method (CE+Ours) consistently outperforms the baseline cross-entropy model across all key metrics. For instance, in NearOOD, our method achieves a significantly lower FPR@95 of 12.93% compared to 54.58% with the baseline, highlighting its superior ability to minimize false positives. Similarly, in FarOOD, our method shows a substantial improvement in AU-ROC, reaching 98.96%, a notable increase from the baseline's 92.36%. Our approach enhances both OOD detection and confidence calibration. Fig.2 illustrates the qualitative comparison, highlighting the advantages of our proposed method. The top row shows the feature space produced by the baseline Cross Entropy (CE) method, while the bottom row shows the feature space generated by our CE+Ours method. In the baseline, the class clusters are relatively close to the origin, and the OOD datasets are also positioned near the origin, making it challenging to effectively differentiate OOD

Table 4: **AUPR_out on CIFAR-10** with OOD sets. Higher values indicate better performance.

| Alg. | Near-OOD | | | Far-OOD | | | | |
|---|---|---|---|---|---|---|---|---|
| | CIFAR-100 | TIN | Avg. | MNIST | SVHN | Textures | Places365 | Avg. |
| DeVries & Taylor (2018) | 85.22 | 85.78 | 85.50 | 98.71 | 97.26 | 81.78 | 96.16 | 93.48 |
| Hendrycks et al. (2019b) | 89.20 | 91.75 | 90.47 | 99.43 | 99.33 | 94.10 | 97.30 | 97.54 |
| Hsu et al. (2020) | 88.09 | 88.41 | 88.25 | 99.83 | 99.12 | 93.41 | 97.05 | 97.35 |
| Tack et al. (2020) | 85.45 | 87.30 | 86.37 | 98.49 | 97.75 | 82.99 | 96.38 | 93.90 |
| Chen et al. (2021) | 83.25 | 82.66 | 82.96 | 98.53 | 93.27 | 78.27 | 95.58 | 91.41 |
| Huang & Li (2021) | 72.70 | 72.13 | 72.41 | 95.24 | 87.20 | 62.67 | 95.85 | 85.24 |
| Du et al. (2022) | 86.17 | 86.98 | 86.57 | 98.53 | 96.37 | 84.05 | 96.83 | 93.95 |
| Wei et al. (2022) | 89.62 | 91.61 | 90.62 | 99.87 | 99.31 | 91.68 | 98.29 | 97.29 |
| Ming et al. (2023) | 87.27 | 88.68 | 87.97 | 98.75 | 99.19 | 88.93 | 97.88 | 96.19 |
| Tao et al. (2023) | 85.63 | 87.08 | 86.36 | 98.54 | 99.60 | 90.18 | 96.48 | 96.20 |
| Regmi et al. (2024a) | 90.20 | 91.91 | 91.06 | 99.89 | 99.54 | 92.85 | 98.17 | 97.61 |
| Hendrycks et al. (2019a) | 89.83 | 98.68 | 94.25 | 97.20 | 99.80 | 96.11 | 98.90 | 98.00 |
| Yu & Aizawa (2019) | 87.07 | 88.38 | 87.73 | 97.01 | 96.39 | 87.12 | 97.30 | 94.45 |
| Yang et al. (2021) | 86.31 | 87.48 | 86.89 | 99.26 | 96.86 | 87.68 | 97.12 | 95.23 |
| Zhang et al. (2023) | 86.88 | 87.90 | 87.39 | 98.63 | 97.06 | 87.48 | 96.95 | 95.03 |
| Cubuk et al. (2020) | 87.07 | 87.59 | 87.33 | 99.13 | 96.65 | 84.24 | 96.85 | 94.22 |
| Hendrycks et al. (2020) | 86.36 | 86.59 | 86.48 | 98.54 | 96.16 | 85.37 | 96.56 | 94.15 |
| Hendrycks et al. (2022) | 90.86 | 92.65 | 91.76 | 98.47 | 99.18 | 98.27 | 98.09 | 98.50 |
| Pinto et al. (2022) | 84.71 | 85.96 | 85.33 | 99.02 | 94.94 | 81.66 | 96.19 | 92.95 |
| **Ours** | 91.26 | 93.66 | 92.46 | 99.52 | 98.74 | 98.52 | 98.27 | 98.76 |

Table 5: **FPR@95 results on CIFAR-100** with various OOD sets.

| Training | Post-processor | Near-OOD | | | Far-OOD | | | | | ID ACC |
|---|---|---|---|---|---|---|---|---|---|---|
| | | CIFAR-10 | TIN | Avg. | MNIST | SVHN | Textures | Places365 | Avg. | |
| ConfBranch(DeVries & Taylor, 2018) | ConfBranch | 74.56 | 65.86 | 70.21 | 55.95 | 76.01 | 85.43 | 69.90 | 71.82 | 76.59 |
| RotPred(Hendrycks et al., 2019b) | RotPred | 72.00 | 53.17 | 62.58 | 22.77 | 15.64 | 40.03 | 59.56 | 34.50 | 76.03 |
| G-ODIN(Hsu et al., 2020) | G-ODIN | 78.82 | 56.34 | 67.58 | 27.19 | 42.68 | 35.83 | 65.03 | 42.68 | 74.46 |
| CSI(Tack et al., 2020) | MSP | 72.62 | 67.90 | 70.26 | 80.54 | 67.21 | 90.51 | 69.41 | 76.92 | 61.60 |
| ARPL(Chen et al., 2021) | ARPL | 64.84 | 58.27 | 61.56 | 59.12 | 59.76 | 71.66 | 62.01 | 63.14 | 70.70 |
| MOS(Huang & Li, 2021) | MOS | 60.60 | 51.49 | 56.05 | 52.70 | 56.33 | 61.24 | 58.86 | 57.28 | 76.98 |
| VOS(Du et al., 2022) | EBO | 59.23 | 51.89 | 55.56 | 48.56 | 47.23 | 62.55 | 56.44 | 53.70 | 77.20 |
| LogitNorm(Wei et al., 2022) | MSP | 73.88 | 51.89 | 62.89 | 34.12 | 47.52 | 77.38 | 55.44 | 53.61 | 76.34 |
| CIDER(Ming et al., 2023) | KNN | 82.71 | 61.33 | 72.02 | 75.32 | 17.82 | 54.43 | 69.30 | 54.22 | - |
| NPOS(Tao et al., 2023) | KNN | 72.50 | 54.21 | 63.35 | 66.98 | 30.67 | 47.39 | 59.47 | 51.13 | - |
| T2FNorm(Regmi et al., 2024a) | T2FNorm | 67.07 | 49.88 | 58.47 | 39.39 | 44.29 | 66.82 | 54.50 | 51.25 | 76.43 |
| OE(Hendrycks et al., 2019a) | MSP | 61.26 | 0.21 | 30.73 | 53.31 | 51.84 | 55.83 | 58.30 | 54.82 | 76.84 |
| MCD(Yu & Aizawa, 2019) | MCD | 62.65 | 49.10 | 55.88 | 62.78 | 43.71 | 56.89 | 54.17 | 54.39 | 75.83 |
| UDG(Yang et al., 2021) | MSP | 66.40 | 56.43 | 61.42 | 45.14 | 59.67 | 71.33 | 59.85 | 59.00 | 71.54 |
| MixOE(Zhang et al., 2023) | MSP | 61.12 | 49.32 | 55.22 | 59.49 | 73.09 | 66.04 | 56.93 | 63.88 | 75.13 |
| RandAugment(Cubuk et al., 2020) | MSP | 59.24 | 50.86 | 55.05 | 66.73 | 60.50 | 59.04 | 57.22 | 60.87 | 78.16 |
| AugMix(Hendrycks et al., 2020) | MSP | 59.27 | 53.33 | 56.30 | 61.94 | 51.89 | 61.35 | 58.24 | 58.36 | 76.45 |
| PixMix(Hendrycks et al., 2022) | MSP | 62.16 | 51.50 | 56.83 | 70.32 | 30.76 | 37.47 | 55.13 | 48.42 | 77.63 |
| RegMixup(Pinto et al., 2022) | MSP | 62.59 | 49.65 | 56.12 | 56.77 | 55.97 | 59.73 | 57.53 | 57.50 | **79.23** |
| **Ours** | MSP | **50.36** | 45.63 | 48.00 | 30.31 | 35.93 | 39.66 | **46.31** | 38.05 | 78.95 |

samples from in-domain samples. In contrast, our method separates the class clusters further from the origin, increasing the feature norms and improving the identification of OOD data. Unlike the baseline CE method, where OOD samples tend to cluster around specific training sets, our method achieves a more uniform circular distribution for the OOD samples. This enhances OOD detection and results in a more balanced feature space representation.

### 4.2 EXPERIMENTS ON CIFAR-10 AND CIFAR-100

Following our MNIST experiments, we adhere to the OpenOOD benchmark (Yang et al., 2022). For our backbone, we used ResNet18 (He et al., 2016), extracting 512-dimensional embeddings from the layer preceding the classification head. When CIFAR-10 is the in-domain dataset, OpenOOD defines CIFAR-100 and TinyImageNet (TIN) as near-OOD, and MNIST, SVHN, Textures, and Places365 as far-OOD. For CIFAR-100 as the in-domain dataset, CIFAR-10 and TIN serve as near-OOD, while MNIST, SVHN, Textures, and Places365 are considered far-OOD. For CIFAR-10, the hyperparameters were set to $\alpha = 1.0$ and $\delta = 1.0$; for CIFAR-100, they were set to $\alpha = 0.1$ and $\delta = 1.0$.

Since the compared methods train models directly, both OOD detection performance and classification accuracy are important metrics. The OOD detection results, measured by FPR@95, AUROC, and AUPR_out, for CIFAR-10 as the in-domain dataset are presented in Tables 2, 3, and 4. Due to space constraints, the values for the "Training" and "Post-processor" columns, as well as the classification accuracy (which are identical across metrics), are omitted in Table 3 and 4. Similar results for CIFAR-100 as the in-domain dataset are provided in Tables 5, 6, and 7.

Our experimental results reveal significant performance variation across different OOD datasets, underscoring the importance of robust average metrics. For the CIFAR-10 experiments, Table 2 shows that our proposed method achieves the second-best near-OOD average FPR@95 and the best far-OOD average FPR@95. In Table 3, our method attains the second-best AUROC in the near-OOD average and the best AUROC in the far-OOD average. Table 4 indicates that for AUPR_out, our method produces the best near-OOD average and the second-best far-OOD average performance. For the CIFAR-100 experiments, Table 5 demonstrates that our approach yields the second-best

Table 6: **AUROC results on CIFAR-100** with OOD sets.

| Alg. | Near-OOD | | | Far-OOD | | | | |
|---|---|---|---|---|---|---|---|---|
| | CIFAR-10 | TIN | *Avg.* | MNIST | SVHN | Textures | Places365 | *Avg.* |
| DeVries & Taylor (2018) | 68.80 | 74.41 | 71.60 | 74.29 | 65.51 | 65.39 | 70.42 | 68.90 |
| Hendrycks et al. (2019b) | 71.11 | 81.75 | 76.43 | **93.10** | 95.39 | 88.16 | 76.95 | **88.40** |
| Hsu et al. (2020) | 73.04 | 81.26 | 77.15 | 91.15 | 83.74 | 89.62 | 78.17 | 85.67 |
| Tack et al. (2020) | 69.50 | 73.40 | 71.45 | 51.79 | 80.24 | 62.22 | 70.99 | 66.31 |
| Chen et al. (2021) | 73.38 | 76.50 | 74.94 | 73.77 | 76.45 | 69.93 | 74.62 | 73.69 |
| Huang & Li (2021) | 78.54 | 82.26 | 80.40 | 80.68 | 81.59 | 79.92 | 78.50 | 80.17 |
| Du et al. (2022) | **79.14** | 82.73 | 80.93 | 82.29 | 84.23 | 78.41 | 80.34 | 81.32 |
| Wei et al. (2022) | 74.57 | 82.37 | 78.47 | 90.69 | 82.80 | 72.37 | 80.25 | 81.53 |
| Ming et al. (2023) | 67.55 | 78.65 | 73.10 | 68.14 | **97.17** | 82.21 | 74.43 | 80.49 |
| Tao et al. (2023) | 75.37 | 81.32 | 78.35 | 73.26 | 92.43 | 85.55 | 77.92 | 82.29 |
| Regmi et al. (2024a) | 76.09 | 83.59 | 79.84 | 86.22 | 86.04 | 77.32 | 81.35 | 82.73 |
| Hendrycks et al. (2019a) | 76.70 | **99.89** | **88.30** | 80.68 | 84.37 | 82.18 | 78.39 | 81.41 |
| Yu & Aizawa (2019) | 75.40 | 78.75 | 77.07 | 68.25 | 75.92 | 77.07 | 77.65 | 74.72 |
| Yang et al. (2021) | 75.15 | 80.90 | 78.02 | 83.88 | 79.80 | 75.57 | 79.11 | 79.59 |
| Zhang et al. (2023) | 78.17 | 83.73 | 80.95 | 76.06 | 72.28 | 77.34 | 79.92 | 76.40 |
| Cubuk et al. (2020) | 78.64 | 81.90 | 80.27 | 69.52 | 76.06 | 78.08 | 78.97 | 75.66 |
| Hendrycks et al. (2020) | 77.80 | 80.91 | 79.36 | 72.75 | 81.16 | 76.32 | 78.51 | 77.18 |
| Hendrycks et al. (2022) | 76.56 | 83.16 | 79.86 | 69.56 | 93.43 | **91.81** | 81.44 | 84.06 |
| Pinto et al. (2022) | 78.40 | 83.25 | 80.83 | 78.75 | 79.47 | 78.13 | 79.79 | 79.04 |
| **Ours** | 78.96 | 83.82 | 81.39 | 90.98 | 87.63 | 86.38 | **85.26** | 87.56 |

Table 7: **AUPR_out results on CIFAR-100** with OOD sets.

| Alg. | Near-OOD | | | Far-OOD | | | | |
|---|---|---|---|---|---|---|---|---|
| | CIFAR-10 | TIN | *Avg.* | MNIST | SVHN | Textures | Places365 | *Avg.* |
| DeVries & Taylor (2018) | 65.88 | 61.93 | 63.90 | 93.34 | 81.02 | 51.74 | 87.24 | 78.33 |
| Hendrycks et al. (2019b) | 66.57 | 70.53 | 68.55 | 98.38 | 97.60 | 78.09 | 89.86 | 90.98 |
| Hsu et al. (2020) | 71.57 | 71.28 | 71.42 | 98.20 | 91.05 | 82.72 | 91.41 | 90.84 |
| Tack et al. (2020) | 67.57 | 62.72 | 65.14 | 88.11 | 91.76 | 51.46 | 88.16 | 79.87 |
| Chen et al. (2021) | 70.67 | 65.05 | 67.86 | 94.24 | 87.95 | 53.85 | 89.49 | 81.38 |
| Huang & Li (2021) | 75.65 | 71.89 | 73.77 | 95.79 | 90.85 | 68.00 | 91.02 | 86.41 |
| Du et al. (2022) | 76.39 | 73.08 | 74.74 | 96.30 | 91.74 | 65.21 | 91.97 | 86.31 |
| Wei et al. (2022) | 72.74 | 72.59 | 72.66 | 98.36 | 90.93 | 57.83 | 91.96 | 84.77 |
| Ming et al. (2023) | 65.66 | 68.96 | 67.31 | 93.71 | **99.02** | 73.11 | 90.11 | 88.99 |
| Tao et al. (2023) | 73.56 | 71.12 | 72.34 | 94.93 | 96.92 | 77.12 | 90.92 | 89.97 |
| Regmi et al. (2024a) | 73.38 | 74.02 | 73.70 | 97.13 | 92.87 | 64.15 | 92.41 | 86.64 |
| Hendrycks et al. (2019a) | 72.95 | **99.86** | **86.40** | 95.92 | 93.03 | 73.84 | 91.31 | 88.53 |
| Yu & Aizawa (2019) | 70.54 | 62.49 | 66.51 | 90.61 | 82.73 | 58.31 | 89.24 | 80.22 |
| Yang et al. (2021) | 72.52 | 71.16 | 71.84 | 96.72 | 90.09 | 62.93 | 91.75 | 85.37 |
| Zhang et al. (2023) | 75.59 | 74.78 | 75.18 | 94.57 | 86.30 | 64.96 | 91.88 | 84.43 |
| Cubuk et al. (2020) | 76.54 | 72.27 | 74.40 | 93.05 | 87.50 | 64.20 | 91.51 | 84.06 |
| Hendrycks et al. (2020) | 75.29 | 70.64 | 72.96 | 93.84 | 90.21 | 61.26 | 91.26 | 84.14 |
| Hendrycks et al. (2022) | 73.91 | 75.02 | 74.47 | 92.86 | 97.43 | **88.76** | **93.08** | **93.03** |
| Pinto et al. (2022) | 76.36 | 74.11 | 75.24 | 95.76 | 89.03 | 63.73 | 91.92 | 85.11 |
| **Ours** | **78.37** | 75.25 | 76.81 | 98.92 | 93.52 | 81.34 | 91.91 | 91.42 |

performance for both the near-OOD and far-OOD averages. Comparable trends are observed in Tables 6 and 7 for AUROC and AUPR_out.

Overall, these results highlight that performance on individual OOD datasets can vary considerably. For example, OE (Hendrycks et al., 2019a) exhibits considerable variation across different OOD datasets (*e.g.*, FPR@95 of 61 on CIFAR-10) but performs extremely well on others (*e.g.*, 0.21 on TIN). This large variation indicates that evaluating an algorithm's robustness requires examining the average metrics for near-OOD and far-OOD datasets. In this context, our method stands out by consistently achieving the best or second-best average performance across all evaluated metrics (FPR@95, AUROC, and AUPR_out).

This robust performance is attributable to our dynamic, data-driven hyperparameter, which is determined solely by the threshold at which the numerical derivative of the softmax output with respect to the feature norm exceeds a specified value ($\delta$) and by the balance factor ($\alpha$) between the ID and OOD losses. Unlike other methods that rely on manually tuned hyperparameters or additional OOD data, our approach automatically adapts to the distribution of the ID data, leading to more consistent and robust OOD detection.

## 5 CONCLUSION

We introduced a method for OOD detection that directly optimizes the feature space. Our approach leverages feature norm optimization to ensure that ID and OOD samples are well-separated in the embedding space. By defining two key thresholds, we derive a scaling ratio that guides the augmentation of OOD samples. These augmented OOD samples, generated by random sampling within a controlled, low-output region, effectively approximate the unknown domain. Our theoretical analysis demonstrates that the risk difference between the true data distribution (*i.e.*, the overall distribution containing both known and unknown samples) and the auxiliary domain constructed using these augmented OOD samples is bounded. This result validates our approach of minimizing risks in the auxiliary domain as an effective surrogate for minimizing risks in the original domain. Extensive experiments confirm that our method significantly outperforms the baseline cross-entropy method.

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

## A PROOF OF THEOREM 1

**Risk Difference Decomposition:** Since by assumption $R_{P,k}(h) = R_{Q,k}(h)$, the overall risk difference is determined solely by the unknown risk:

$$\left| R_P^\alpha(h) - R_Q^\alpha(h) \right| = \left| (1-\alpha)R_{P,k}(h) + \alpha R_{P,u}(h) - \left[ (1-\alpha)R_{P,k}(h) + \alpha R_{Q,u}(h) \right] \right|$$
$$= \alpha \left| R_{P,u}(h) - R_{Q,u}(h) \right|. \tag{A.1}$$

We now focus on bounding $\Delta := |R_{P,u}(h) - R_{Q,u}(h)|$.

**Expressing the Difference in Unknown Risks:** By definition,

$$R_{P,u}(h) = \int_{\mathcal{X}} \ell\Big( h(g(x)), \bar{y} \Big) \, dP_{X|Y \in \mathcal{Y}_u}(x) \tag{A.2}$$

and

$$R_{Q,u}(h) = \int_{\mathcal{X}} \ell\Big( h(g(x)), \bar{y} \Big) \, dQ_{X|Y \in \mathcal{Y}_u}(x). \tag{A.3}$$

Define the signed measure

$$\mu(x) = \big( P_{X|Y \in \mathcal{Y}_u} - Q_{X|Y \in \mathcal{Y}_u} \big)(x). \tag{A.4}$$

Then,

$$R_{P,u}(h) - R_{Q,u}(h) = \int_{\mathcal{X}} \ell\Big( h(g(x)), \bar{y} \Big) \, d\mu(x). \tag{A.5}$$

Taking the absolute value,

$$|R_{P,u}(h) - R_{Q,u}(h)| = \left| \int_{\mathcal{X}} \ell\Big( h(g(x)), \bar{y} \Big) \, d\mu(x) \right|. \tag{A.6}$$

**Decomposing the Integral via an Intermediate Function:** Let $h' \in \mathcal{H}$ be arbitrary. Then, add and subtract $\ell\Big( h'(g(x)), \bar{y} \Big)$ inside the integral:

$$R_{P,u}(h) - R_{Q,u}(h) = \int_{\mathcal{X}} \Big[ \ell\Big( h(g(x)), \bar{y} \Big) - \ell\Big( h'(g(x)), \bar{y} \Big) \Big] d\mu(x)$$
$$+ \int_{\mathcal{X}} \ell\Big( h'(g(x)), \bar{y} \Big) d\mu(x). \tag{A.7}$$

Taking absolute values and applying the triangle inequality for integrals, we obtain

$$|R_{P,u}(h) - R_{Q,u}(h)| \leq I_1 + I_2, \tag{A.8}$$

where

$$I_1 := \left| \int_{\mathcal{X}} \Big[ \ell\Big( h(g(x)), \bar{y} \Big) - \ell\Big( h'(g(x)), \bar{y} \Big) \Big] d\mu(x) \right| \tag{A.9}$$

and

$$I_2 := \left| \int_{\mathcal{X}} \ell\Big( h'(g(x)), \bar{y} \Big) d\mu(x) \right|. \tag{A.10}$$

**Bounding $I_1$ Using the Triangle Inequality:** By the triangle inequality for the loss $\ell$ (which satisfies the triangle inequality), for any $x$ we have

$$\left| \ell\Big( h(g(x)), \bar{y} \Big) - \ell\Big( h'(g(x)), \bar{y} \Big) \right| \leq \ell\Big( h(g(x)), h'(g(x)) \Big). \tag{A.11}$$

Therefore,

$$I_1 \leq \int_{\mathcal{X}} \ell\Big( h(g(x)), h'(g(x)) \Big) \, d|\mu|(x), \tag{A.12}$$

where $d|\mu|(x)$ denotes the total variation measure of $\mu$.

**Relating the First Term to $\Lambda$:** We now wish to bound

$$\int_{\mathcal{X}} \ell\Big(h(g(x)), h'(g(x))\Big) \, d|\mu|(x). \tag{A.13}$$

There are $h'$ that satisfies

$$\ell\Big(h(g(x)), h'(g(x))\Big) \le \ell\Big(h'(g(x)), \bar{y}\Big). \tag{A.14}$$

Let us define the hypothesis space $\mathcal{H}'$ for $h'$ such that it satisfies equation A.14. Then, for any $h' \in \mathcal{H}'$, integrating both sides with respect to the positive measure $P_{X|Y \in \mathcal{Y}_u} + Q_{X|Y \in \mathcal{Y}_u}$, we obtain

$$\int_{\mathcal{X}} \ell\Big(h(g(x)), h'(g(x))\Big) \, d\Big(P_{X|Y \in \mathcal{Y}_u} + Q_{X|Y \in \mathcal{Y}_u}\Big)(x) \le \int_{\mathcal{X}} \ell\Big(h'(g(x)), \bar{y}\Big) \, d\Big(P_{X|Y \in \mathcal{Y}_u} + Q_{X|Y \in \mathcal{Y}_u}\Big)(x). \tag{A.15}$$

The right-hand side is equal to $R_{P,u}(h') + R_{Q,u}(h')$. Since this inequality holds for any $h' \in \mathcal{H}'$, we define the residual term as

$$\Lambda' := \min_{h' \in \mathcal{H}'} \Big(R_{P,u}(h') + R_{Q,u}(h')\Big). \tag{A.16}$$

Under our training assumption for unknown samples, the model is encouraged to produce low-confidence predictions. As a result, for unknown $x$, we expect that $h(g(x)) \approx \bar{y}$.

As $h(g(x))$ approaches $\bar{y}$, the hypothesis space $\mathcal{H}'$ expands. When $h(g(x))$ exactly matches $\bar{y}$, the space $\mathcal{H}'$ coincides with $\mathcal{H}$. Consequently, we obtain

$$\min_{h' \in \mathcal{H}'} \Big(R_{P,u}(h') + R_{Q,u}(h')\Big) \approx \min_{h' \in \mathcal{H}} \Big(R_{P,u}(h') + R_{Q,u}(h')\Big) := \Lambda. $$

Thus, we conclude that $I_1 \le \Lambda$.

**Bounding $I_2$ by the Disparity Discrepancy Metric:** The second term,

$$I_2 = \left| \int_{\mathcal{X}} \ell\Big(h'(g(x)), \bar{y}\Big) \, d\mu(x) \right|, \tag{A.17}$$

is bounded by definition of the disparity discrepancy metric:

$$I_2 \le \sup_{h' \in \mathcal{H}} \left| \int_{\mathcal{X}} \ell\Big(h'(g(x)), \bar{y}\Big) \, d\Big(P_{X|Y \in \mathcal{Y}_u} - Q_{X|Y \in \mathcal{Y}_u}\Big)(x) \right| = d^l_{h,\mathcal{H}}\Big(P_{X|Y \in \mathcal{Y}_u}, Q_{X|Y \in \mathcal{Y}_u}\Big). \tag{A.18}$$

**Final Bound:** Combining the bounds for $I_1$ and $I_2$, we obtain

$$|R_{P,u}(h) - R_{Q,u}(h)| \le \Lambda + d^l_{h,\mathcal{H}}\Big(P_{X|Y \in \mathcal{Y}_u}, Q_{X|Y \in \mathcal{Y}_u}\Big). \tag{A.19}$$

Finally, since

$$\big|R_P^\alpha(h) - R_Q^\alpha(h)\big| = \alpha \, |R_{P,u}(h) - R_{Q,u}(h)|, \tag{A.20}$$

it follows that

$$\big|R_P^\alpha(h) - R_Q^\alpha(h)\big| \le \alpha \, d^l_{h,\mathcal{H}}\Big(P_{X|Y \in \mathcal{Y}_u}, Q_{X|Y \in \mathcal{Y}_u}\Big) + \alpha \, \Lambda. \tag{A.21}$$

Although we have established that $\big|R_P^\alpha(h) - R_Q^\alpha(h)\big|$ is bounded, we must further demonstrate the relationship between the minimizers of the two risks. Specifically, we prove that

$$\min_{h \in \mathcal{H}} R_Q^\alpha(h) = \min_{h \in \mathcal{H}} R_P^\alpha(h) \tag{A.22}$$

and

$$\arg\min_{h \in \mathcal{H}} R_Q^\alpha(h) \subset \arg\min_{h \in \mathcal{H}} R_P^\alpha(h). \tag{A.23}$$

**Assumption:** Following Assumption 1 in Fang et al. (2021), we assume that there exists a hypothesis $\tilde{h} \in \mathcal{H}$ and a distribution $\tilde{P}$ defined over $\mathcal{X}$, with $\mathrm{supp}(\tilde{P}) = \mathcal{X}$, such that

$$\int_{\mathcal{X} \times \mathcal{Y}} l\big(\phi \circ \tilde{h}(g(x)), \phi(y)\big) dP_{Y|X}(y|x) d\tilde{P}(x) = 0, \tag{A.24}$$

where $\phi$ is a function on $\mathcal{Y}$ defined by

$\phi(y) = \begin{cases} \bar{y}, & \text{if } y \in \mathcal{Y}_u, \\ y_1, & \text{otherwise.} \end{cases}$ In other words, the hypothesis space $\mathcal{H}$ is sufficiently expressive so that there exist hypotheses that can perfectly classify the unknown classes.

**Consistency of Minimum $\alpha$-Risk:** By the above assumption, there exists some $h \in \mathcal{H}$ such that $R_{P,u}(h) = 0$; similarly, there exists some $h \in \mathcal{H}$ such that $R_{Q,u}(h) = 0$. Hence, we have

$$\min_{h \in \mathcal{H}} R_P^\alpha(h) \tag{A.25}$$

$$= \min_{h \in \mathcal{H}} (1 - \alpha) \int_{\mathcal{X} \times \mathcal{Y}_k} l(h(g(x)), y) dP_{X|Y \in \mathcal{Y}_k}(x, y) + \alpha \int_{\mathcal{X}} l(h(g(x)), \bar{y}) dP_{X|Y \in \mathcal{Y}_u}(x) \tag{A.26}$$

$$= (1 - \alpha) \min_{h \in \mathcal{H}} R_{P,k}(h), \tag{A.27}$$

Then similar to $R_P^\alpha$, $R_Q^\alpha$ follows,

$$\min_{h \in \mathcal{H}} R_Q^\alpha(h) = (1 - \alpha) \min_{h \in \mathcal{H}} R_{Q,k}(h). \tag{A.28}$$

Moreover, since $P_{X|Y \in \mathcal{Y}k} = Q_{X|Y \in \mathcal{Y}k}$, it follows that

$$\min_{h \in \mathcal{H}} R_{P,k}^\alpha(h) = \min_{h \in \mathcal{H}} R_{Q,k}^\alpha(h), \tag{A.29}$$

and thus,

$$\min_{h \in \mathcal{H}} R_P^\alpha(h) = (1 - \alpha) \min_{h \in \mathcal{H}} R_{P,k}^\alpha(h) = (1 - \alpha) \min_{h \in \mathcal{H}} R_{Q,k}^\alpha(h) = \min_{h \in \mathcal{H}} R_Q^\alpha(h). \tag{A.30}$$

This shows that the minimum value of the $\alpha$-risk for distribution P is equal to the minimum value of the $\alpha$-risk for distribution Q .

**Optimal Hypothesis Consistency Between $Q$ and $P$:** Let $h_Q \in \arg\min_{h \in \mathcal{H}} R_Q^\alpha(h)$ be any optimal hypothesis for $R_Q^\alpha$. Since $R_{Q,u}(h_Q) = 0$, we have

$$R_Q^\alpha(h_Q) = (1 - \alpha) R_{Q,k}(h_Q) = (1 - \alpha) R_{P,k}(h_Q). \tag{A.31}$$

Moreover, since $P_X \ll Q_X$, it follows that $P_{X|Y \in \mathcal{Y}_u} \ll Q_{X|Y \in \mathcal{Y}_u}$.

Therefore,

$$\int_{\mathcal{X}} l\big(h_Q(g(x)), \bar{y}\big) dQ_{X|Y \in \mathcal{Y}_u}(x) = \int_{\mathcal{X}} l\big(h_Q(g(x)), \bar{y}\big) dP_{X|Y \in \mathcal{Y}_u}(x) = 0, \tag{A.32}$$

which implies that

$$R_Q^\alpha(h_Q) = (1 - \alpha) R_{Q,k}(h_Q) = (1 - \alpha) R_{P,k}(h_Q) = R_P^\alpha(h_Q). \tag{A.33}$$

Furthermore, by the consistency of the minimum $\alpha$-risk,

$$R_Q^\alpha(h_Q) = \min_{h \in \mathcal{H}} R_Q^\alpha(h) = \min_{h \in \mathcal{H}} R_P^\alpha(h). \tag{A.34}$$

Thus, any hypothesis that minimizes $R_Q^\alpha(h)$ also minimizes $R_P^\alpha(h)$. This implies that

$$\arg\min_{h \in \mathcal{H}} R_Q^\alpha(h) \subset \arg\min_{h \in \mathcal{H}} R_P^\alpha(h). \tag{A.35}$$

**Algorithm 1** Feature Norm-based OOD Augmentation

---

**Require:** Network $g(\cdot)$, ID dataset $\mathcal{D}_{\text{ID}}$, class centers $C = \{c_1, \ldots, c_K\}$, derivative threshold $\delta$, batch size $B$, OOD loss weight $\lambda$

**Ensure:** Trained network parameters

1: **for** each training iteration **do**
2:     Sample mini-batch $\{(x_i, y_i)\}_{i=1}^{B} \sim \mathcal{D}_{\text{ID}}$
3:     Compute embeddings and norms: $z_i \leftarrow g(x_i),\ n_i \leftarrow \|z_i\|$
4:     Compute baseline norm

$$N_0 \leftarrow \arg\min_{n} \left\| \text{softmax}\big(C^{\top}(z_i \tfrac{n}{n_i})\big) - \tfrac{1}{K}\mathbf{1} \right\|$$

5:     Compute saturation bound

$$N_{\text{cap}} \leftarrow \min\Big\{ n > N_0 : \tfrac{d}{dn} \max_{k} \big[\text{softmax}(C^{\top}(z_i \tfrac{n}{n_i}))\big]_k > \delta \Big\}$$

6:     Compute scaling ratio and mean norm: $\tau \leftarrow \frac{N_0}{N_{\text{cap}}},\ \mu_{\text{ID}} \leftarrow \frac{1}{B}\sum_{i=1}^{B} n_i$
7:     Generate OOD embeddings $\{\tilde{z}_i\}$ s.t. $\|\tilde{z}_i\| \leq \tau\,\mu_{\text{ID}}$
8:     Compute losses:

$$\mathcal{L}_{\text{ID}} \leftarrow \tfrac{1}{B}\sum_{i=1}^{B} \text{CE}\big(\text{softmax}(Wz_i),\, y_i\big), \quad \mathcal{L}_{\text{OOD}} \leftarrow \tfrac{1}{B}\sum_{i=1}^{B} \max_{k}\big[\text{softmax}(W\tilde{z}_i)\big]_k$$

9:     Total loss and update: $\mathcal{L} \leftarrow \mathcal{L}_{\text{ID}} + \lambda\,\mathcal{L}_{\text{OOD}}$
10:    Update parameters via $\nabla\mathcal{L}$
11: **end for**

---

# B   Algorithmic Description

Algorithm 1 describes the whole pipeline of the proposed feature-Norm-based OOD augmentation.

# C   Interpretable Hyperparameter Tuning

Our central hypothesis—that aligning feature norm with confidence enhances OOD detection—also drives interpretable hyperparameter tuning: We selected $\delta$ based on the separability of feature norms between ID and OOD samples (Fig.1, left). For $\alpha$, which balances ID and OOD loss, we prioritized maintaining ID classification accuracy, as very small $\alpha$ can excessively lower ID confidence and hurt accuracy. We believe that this interpretability-driven hyperparameter selection, based OOD metrics, feature norm distributions, and ID accuracy, makes our method more robust and explainable.

# D   Experimental results on ImageNet-200

We also conducted OOD detection experiments on ImageNet-200. As shown in Tables A.1–A.3, our method achieved consistently strong performance across both near- and far-OOD settings, ranking 3rd/2nd/3rd on FPR@95 (near/far/ID), 2nd on AUROC (near/far), and 1st/2nd on AUPR-out (near/far). This contrasts with many methods that excel on only one type of OOD.

Table A.1: **FPR@95 results on ImageNet-200** with various OODs. Please refer to the main text for references.

| Alg. | Near-OOD | | | Far-OOD | | | | ID ACC |
|---|---|---|---|---|---|---|---|---|
| | SSB-hard | NINCO | *Avg.* | iNaturalist | Textures | OpenImage-O | *Avg.* | |
| ConfBranch(DeVries & Taylor, 2018) | 72.24 | 50.63 | 61.44 | 23.84 | 42.42 | 37.99 | 34.75 | 85.92 |
| CRotPred(Hendrycks et al., 2019b) | 72.00 | 48.84 | 60.42 | 20.51 | 26.44 | 31.51 | 26.16 | 86.37 |
| G-ODIN(Hsu et al., 2020) | 78.23 | 61.52 | 69.87 | 26.13 | 28.98 | 35.43 | 30.18 | 84.56 |
| ARPL(Chen et al., 2021) | 65.73 | 45.75 | 55.74 | 29.32 | 42.87 | 37.20 | 36.46 | 83.95 |
| MOS(Huang & Li, 2021) | 74.35 | 68.85 | 71.60 | 49.55 | 51.27 | 53.86 | 51.56 | 85.60 |
| VOS(Du et al., 2022) | 69.93 | 49.85 | 59.89 | 25.53 | 39.74 | 36.77 | 34.01 | 86.23 |
| LogitNorm(Wei et al., 2022) | 67.46 | 45.46 | 56.46 | 15.70 | 32.13 | 30.49 | 26.11 | 86.04 |
| CIDER(Ming et al., 2023) | 75.50 | 44.69 | 60.10 | 26.54 | 31.51 | 32.47 | 30.17 | / |
| NPOS(Tao et al., 2023) | 74.29 | 49.89 | 62.09 | 20.01 | 16.87 | 28.40 | **21.76** | / |
| T2FNorm(Regmi et al., 2024a) | 65.94 | 44.09 | 55.01 | 13.47 | 33.46 | 29.17 | 25.37 | 86.87 |
| OE(Hendrycks et al., 2019a) | 64.67 | 39.93 | **52.30** | 27.03 | 41.92 | 33.56 | 34.17 | 85.82 |
| MCD(Yu & Aizawa, 2019) | 65.69 | 43.74 | 54.71 | 21.74 | 38.11 | 29.93 | 29.93 | 86.12 |
| UDG(Yang et al., 2021) | 75.84 | 61.94 | 68.89 | 49.26 | 71.94 | 64.92 | 62.04 | 68.11 |
| MixOE(Zhang et al., 2023) | 68.26 | 47.69 | 57.97 | 30.84 | 51.44 | 40.51 | 40.93 | 85.71 |
| RandAugment(Cubuk et al., 2020) | 65.97 | 44.39 | 55.18 | 25.82 | 44.16 | 35.41 | 35.13 | 86.58 |
| AugMix(Hendrycks et al., 2020) | 65.91 | 44.02 | 54.97 | 25.08 | 41.49 | 33.70 | 33.42 | 87.01 |
| PixMix(Hendrycks et al., 2022) | 67.81 | 46.27 | 57.04 | 27.29 | 44.42 | 37.07 | 36.26 | 85.79 |
| RegMixup(Pinto et al., 2022) | 65.70 | 41.85 | 53.78 | 24.70 | 42.20 | 34.73 | 33.88 | **87.25** |
| **Ours** | 65.56 | 43.95 | 54.76 | 14.02 | 29.63 | 22.16 | 21.94 | 86.98 |

Table A.2: **AUROC results on ImageNet-200** with various OODs.

| Alg. | Near-OOD | | | Far-OOD | | | |
|---|---|---|---|---|---|---|---|
| | SSB-hard | NINCO | *Avg.* | iNaturalist | Textures | OpenImage-O | *Avg.* |
| ConfBranch(DeVries & Taylor, 2018) | 75.01 | 83.19 | 79.10 | 93.40 | 89.64 | 88.26 | 90.43 |
| CRotPred(Hendrycks et al., 2019b) | 77.04 | 86.15 | 81.59 | 93.47 | 93.81 | 90.41 | 92.56 |
| G-ODIN(Hsu et al., 2020) | 72.94 | 81.63 | 77.28 | 93.12 | 93.67 | 90.18 | 92.33 |
| ARPL(Chen et al., 2021) | 79.24 | 84.81 | 82.02 | 91.54 | 88.11 | 88.04 | 89.23 |
| MOS(Huang & Li, 2021) | 66.54 | 73.14 | 69.84 | 79.69 | 81.38 | 80.29 | 80.46 |
| VOS(Du et al., 2022) | 79.68 | 85.35 | 82.51 | 92.77 | 90.95 | 89.28 | 91.00 |
| LogitNorm(Wei et al., 2022) | 78.42 | 86.90 | 82.66 | 96.26 | 91.85 | 91.01 | 93.04 |
| CIDER(Ming et al., 2023) | 76.04 | 85.13 | 80.58 | 90.69 | 92.38 | 88.92 | 90.66 |
| NPOS(Tao et al., 2023) | 74.29 | 84.50 | 79.40 | 94.81 | 96.97 | 91.69 | **94.49** |
| T2FNorm(Regmi et al., 2024a) | 79.00 | 86.99 | 83.00 | 96.87 | 91.95 | 91.81 | 93.55 |
| OE(Hendrycks et al., 2019a) | 82.34 | 87.35 | **84.84** | 90.30 | 87.76 | 89.01 | 89.02 |
| MCD(Yu & Aizawa, 2019) | 81.51 | 85.74 | 83.62 | 90.83 | 86.87 | 89.12 | 88.94 |
| UDG(Yang et al., 2021) | 70.73 | 77.88 | 74.30 | 85.95 | 81.79 | 78.54 | 82.09 |
| MixOE(Zhang et al., 2023) | 80.23 | 85.01 | 82.62 | 90.64 | 86.80 | 87.36 | 88.27 |
| RandAugment(Cubuk et al., 2020) | 80.18 | 86.16 | 83.17 | 93.07 | 88.81 | 89.12 | 90.34 |
| AugMix(Hendrycks et al., 2020) | 80.43 | 86.55 | 83.49 | 93.17 | 89.28 | 89.61 | 90.68 |
| PixMix(Hendrycks et al., 2022) | 78.79 | 85.51 | 82.15 | 92.48 | 89.81 | 88.41 | 90.23 |
| RegMixup(Pinto et al., 2022) | 80.85 | 87.41 | 84.13 | 93.28 | 89.59 | 89.56 | 90.81 |
| **Ours** | 81.23 | 87.36 | 84.30 | 96.23 | 92.22 | 91.98 | 94.02 |

Table A.3: **AUPR_out results on ImageNet-200** with various OODs.

| Alg. | Near-OOD | | | Far-OOD | | | |
|---|---|---|---|---|---|---|---|
| | SSB-hard | NINCO | *Avg.* | iNaturalist | Textures | OpenImage-O | *Avg.* |
| ConfBranch(DeVries & Taylor, 2018) | 92.94 | 71.27 | 82.11 | 92.18 | 83.05 | 90.80 | 88.67 |
| CRotPred(Hendrycks et al., 2019b) | 93.23 | 76.51 | 84.87 | 90.85 | 87.33 | 92.11 | 90.10 |
| G-ODIN(Hsu et al., 2020) | 92.42 | 73.12 | 82.77 | 92.81 | 90.23 | 93.07 | 92.04 |
| ARPL(Chen et al., 2021) | 94.36 | 74.33 | 84.35 | 90.97 | 80.84 | 91.09 | 87.63 |
| MOS(Huang & Li, 2021) | 89.15 | 57.62 | 73.38 | 73.75 | 61.76 | 82.86 | 72.79 |
| VOS(Du et al., 2022) | 94.72 | 76.47 | 85.59 | 91.69 | 86.42 | 92.24 | 90.11 |
| LogitNorm(Wei et al., 2022) | 94.05 | 78.76 | 86.41 | 96.16 | 87.10 | 93.48 | 92.25 |
| CIDER(Ming et al., 2023) | 93.22 | 73.41 | 83.32 | 88.09 | 88.15 | 91.26 | 89.16 |
| NPOS(Tao et al., 2023) | 92.88 | 75.86 | 84.37 | 94.66 | 95.63 | 94.21 | **94.83** |
| T2FNorm(Regmi et al., 2024a) | 94.26 | 79.03 | 86.64 | 96.82 | 87.57 | 94.21 | 92.87 |
| OE(Hendrycks et al., 2019a) | 95.59 | 78.12 | 86.86 | 86.42 | 77.91 | 91.11 | 85.15 |
| MCD(Yu & Aizawa, 2019) | 95.13 | 73.75 | 84.44 | 87.01 | 71.21 | 90.47 | 82.90 |
| UDG(Yang et al., 2021) | 91.35 | 64.82 | 78.09 | 85.54 | 74.60 | 84.74 | 81.63 |
| MixOE(Zhang et al., 2023) | 94.83 | 74.72 | 84.78 | 88.94 | 78.66 | 90.43 | 86.01 |
| RandAugment(Cubuk et al., 2020) | 94.74 | 76.67 | 85.71 | 92.44 | 82.74 | 91.95 | 89.04 |
| AugMix(Hendrycks et al., 2020) | 94.79 | 76.94 | 85.86 | 92.28 | 83.14 | 92.20 | 89.21 |
| PixMix(Hendrycks et al., 2022) | 94.19 | 75.62 | 84.90 | 91.62 | 85.39 | 91.50 | 89.50 |
| RegMixup(Pinto et al., 2022) | 94.90 | 78.69 | 86.80 | 92.57 | 83.77 | 92.28 | 89.54 |
| **Ours** | 94.89 | 78.95 | **86.92** | 96.12 | 93.28 | 94.85 | 94.75 |

