# OpenReview forum: "Fitting Feature Norm to Confidence: A Regularization Approach for Robust Out-of-Distribution Detection"
_ICLR.cc/2026/Conference — ICLR 2026 Conference Withdrawn Submission_

### Official Review · Reviewer_6EPJ · 2025-10-22

**Soundness:** 1
**Presentation:** 3
**Contribution:** 2
**Rating:** 2
**Confidence:** 3

**Summary:**

This paper introduces a novel approach to out-of-distribution (OOD) detection that eliminates the need for explicit OOD samples during training. The method leverages the intrinsic relationship between feature norm and model confidence to generate synthetic OOD examples directly in the embedding space. Two feature-norm thresholds, computed from in-domain (ID) data on a batch basis, define a controlled confidence range. For each batch, the average ID feature norm $\mu_{ID}$ is measured, and pseudo-OOD embeddings are uniformly sampled within a ball of radius $\tau\mu_{ID}$, where $\tau$ is the ratio between the zero-confidence and confidence-saturation thresholds. These synthetic points represent low-confidence regions of the latent space. Training is performed using a standard cross-entropy loss on ID samples, combined with a regularization term that penalizes overconfident predictions on the augmented OOD embeddings.

**Strengths:**

1.	The method does not require access to explicit OOD samples, as these are artificially generated within the embedding space during training.
2.	The method is computationally inexpensive.
3.	A theoretical analysis is provided.
4.	The method mitigates the overconfidence of OOD samples..

**Weaknesses:**

1.	The novelty should be clarified, particularly in comparison with Fang et al. (2021).
2.	The methods relies on strong assumptions.
3.	Restricted architectural evaluation
4.	The method requires access to the training set or to labelled InD data.
5.	Fairness concern in tuning the parameter $\delta$

**Questions:**

1. The proposed method is inspired by Fang et al. (2021); however, the precise contribution relative to their work is not clearly stated. The authors should clarify how their approach advances beyond Fang et al., in particular regarding the controlled alignment between feature norms and model confidence, the generation of synthetic OOD embeddings in a low-confidence region, and the practical training framework combining ID and OOD losses. Clearer differentiation of the novelty would strengthen the paper.
2.	The method relies on the assumption that there is a strong correlation between embedding norm and model confidence. However, this relationship may not hold consistently across different architectures or normalization schemes (see  point #3 below). The authors should clarify more specifically the types of networks or models for which their method is expected to be effective, and discuss potential limitations when this assumption does not hold.
3.	The experiments are conducted solely on LeNet and ResNet models. While LeNet and ResNet are standard benchmarks, evaluating only on these two architectures limits evidence for the method’s robustness and applicability. Networks with different depth, normalization schemes, or embedding structures could behave differently, especially regarding the feature norm–confidence relationship.It remains unclear how the proposed method performs on other architectures, such as deeper or transformer-based networks, which may have different embedding characteristics.
4.	The artificial OOD samples are generated via uniform sampling in the latent space. The authors should provide a clear justification for this choice and discuss its theoretical or empirical rationale. In particular, it is unclear why the center of the sampling ball is set at the origin rather than at the batch mean $\mu_{ID}$ (see point #5 below). Additionally, it would be important to clarify how the method behaves when the feature norms vary significantly across classes, as this could affect the effectiveness of the OOD augmentation and the assumption that low-norm samples correspond to low confidence.
5.	It is unclear whether the proposed method relies on a neural collapse phenomenon, in which the in-domain (ID) features are mapped onto a hypersphere, resulting in roughly equal norms across classes, while OOD samples are positioned near the average ID features. The authors should clarify if their approach depends on this behavior, and discuss how deviations from neural collapse  might affect the effectiveness of the OOD augmentation and the feature-norm–confidence relationship.
6.	The procedure for setting the parameter $\delta$ (used to determine $N_{cap}$) is not clearly described. It is only mentionned « we selected $\delta$ based on the separability of feature norms between ID and OoD samples (Fig 1 left)» which raises serious concerns about fairness. In a proper evaluation setting, test-time or OOD information should not be used to tune hyperparameters. The authors should clarify how $\delta$ is determined in a fair and reproducible manner, indicate whether it is dataset- or architecture-dependent, and analyze the method’s sensitivity to this choice.

This paper presents an interesting and theoretically grounded approach to OOD detection without requiring explicit OOD data. While the idea of leveraging feature norms to generate synthetic OOD samples is appealing and computationally efficient, the paper would benefit from clearer differentiation from prior work, stronger experimental validation across architectures, and a more rigorous justification of parameter choices.

---

### Official Review · Reviewer_ExVJ · 2025-10-27

**Soundness:** 2
**Presentation:** 2
**Contribution:** 2
**Rating:** 2
**Confidence:** 4

**Summary:**

The paper proposes a training-time regularization scheme for OOD detection that explicitly links feature-norm magnitude to classifier confidence. Extensive experiments show that the method significantly enhances OOD detection performance.

**Strengths:**

Generally clear and readable; figures and tables are informative.

**Weaknesses:**

1. lack of experiments on large-scale datasets (e.g., ImageNet-1k) and transformer-based backbones (e.g. ViT)
2. comprasion is not comprehensive. this paper only compares with 1 baselines proposed from 2024
3. Theorem 1 assumes that $\ell$ satisfies the triangle inequality. For standard losses (including any metric), this inequality is not guaranteed. With $\ell$ as cross-entropy, it’s even worse (no triangle inequality at all). what loss is used as $\ell$ in this paper?
4. Eqs. (A.22, A.23) assume that there exists a hypothesis perfectly handling unknowns, which seems to be strong and should be justified. Even the assumption can (unreasonably) hold in practice, Eq. (6) in Theorem 1 addinitionally assumes that the same $h$  (or at least some common $h$) simultaneously achieves the minima for both $P$ and $Q$, which is not clearly stated and justified in this paper.
5. the claim in Lines 671-672 is heuristic and depends on the unknown $h$, which is not reasonable in practice.
6. lack of ablation study on hyperparameters
7. Compared with full density estimation in Fang et al.(2021) that is thorectically justified, the use of uniform distribution is a simplified version in practice. The authors should give thorectical guarantees for the effectiveness of this operation.

**Questions:**

see Weaknesses.

---

### Official Review · Reviewer_PBAQ · 2025-10-31

**Soundness:** 2
**Presentation:** 1
**Contribution:** 2
**Rating:** 2
**Confidence:** 4

**Summary:**

This paper proposed a novel framework to achieve robust OOD detection by explicitly designing the feature space to align feature norms with model confidence. Extensive experimental results demonstrate that the proposed method markedly improves OOD detection performance across multiple datasets.

**Strengths:**

1  OOD detection is important for safety-critic applications of machine learning models.

2 Pseudo-OOD data augmentation at the feature level has been widely validated as effective.

**Weaknesses:**

1. The effectiveness and robustness: As can be seen in the Tab~2, 3, 4, 5 ,the performance improvement of this method is marginal, and it is not effective on all OOD test datasets.
2. The fairness of the comparison:  As is shown in the Table~2 in the main paper, the OOD detection methods using pseudo-OOD data are based on different post-hoc method (shown as "post-processor" in the table) , which is unfair for comparing. In particular, to my understanding, the method in this paper falls into the same category as VOS and NPOS, which involves generating pseudo-data at the feature level for OOD fine-tuning. Consequently, using the same post-hoc method is essential to validate the effectiveness of the generated pseudo-OOD data fairly.
3. Lack of ablation studies: The proposed method involves numerous hyper-parameters in its implementation, yet the paper lacks corresponding ablation studies, which weakens the soundness of the paper.
4. Lack of comprehensive experiments: The experiments were performed with only CNN architecture. And no experiments were carried out on the ImageNet-1K benchmark, which is more challenging for OOD detection.
5. Lack of comprehensive comparison: A number of recent related works have not been compared in the paper, e.g., [1], [2], [3], and [4].
6. The presentation of the manuscript requires improvement: For instance, many tables are missing key information. Tables~A.1-3 in the Appendix, for example, do not specify the model architectures and post-hoc methods employed.

[1] Yeonguk Yu, Sungho Shin, Seongju Lee, Changhyun Jun, and Kyoobin Lee. Block selection method for using feature norm in out-of-distribution detection. In Proceedings of the IEEE/CVF Conference on Computer Vision and Pattern Recognition, pages 15701–15711, 2023.

[2] Haoqi Wang, Zhizhong Li, Litong Feng, and Wayne Zhang.Vim: Out-of-distribution with virtual-logit matching. In Proceedings of the IEEE/CVF Conference on Computer Vision and Pattern Recognition, 2022.

[3] Yue Song, Nicu Sebe, and Wei Wang. Rankfeat: Rank-1 feature removal for out-of-distribution detection. Advances in Neural Information Processing Systems, 35:17885–17898, 2022.

[4] Yue Song, Wei Wang, and Nicu Sebe. Rankfeat&rankweight: Rank-1 feature/weight removal for out-of-distribution detection. IEEE Transactions on Pattern Analysis and Machine Intelligence, 2024.

**Questions:**

Please see the Weaknesses.

---

### Official Review · Reviewer_ZzrK · 2025-10-31

**Soundness:** 2
**Presentation:** 2
**Contribution:** 2
**Rating:** 2
**Confidence:** 4

**Summary:**

this paper proposes a novel framework for robust out-of-distribution (OOD) detection by explicitly designing the feature space.
Extensive experiments show that the proposed approach significantly enhances OOD detection performance.

**Strengths:**

1. this paper is well written
2. the provided pseudo-code is informative

**Weaknesses:**

1. the main theorem is mainly borrowed from Fang et al.(2021), which lacks significant novelty.
2. compared with Fang et al.(2021), this paper uses the uniform distribution, which lacks clear motivation.
3. experiments on imagenet-1k lacks
4. why there is only one 2024 paper used as baseline in this paper

**Questions:**

see weakness

---

### Note · Authors · 2025-11-12

I have read and agree with the venue's withdrawal policy on behalf of myself and my co-authors.